# Continuous Cropping Inhibits Photosynthesis of *Polygonatum odoratum*

**DOI:** 10.3390/plants12193374

**Published:** 2023-09-25

**Authors:** Yan Wang, Yunyun Zhou, Jing Ye, Chenzhong Jin, Yihong Hu

**Affiliations:** 1College of Agriculture and Biotechnology, Hunan University of Humanities, Science and Technology, Loudi 417000, China; 3087@huhst.edu.cn (Y.W.); 3022@huhst.edu.cn (Y.Z.); 8026@huhst.edu.cn (J.Y.); 2Hunan Province Key Laboratory of Plant Functional Genomics and Developmental Regulation, State Key Laboratory of Chemo/Biosensing and Chemometrics, National Center of Technology Innovation for Saline-Alkali Tolerant Rice, College of Biology, Hunan University, Changsha 410082, China; 3Biodiversity Institute, Hunan Academy of Forestry, Changsha 410018, China

**Keywords:** *P. odoratum*, leaf, photosynthesis, carbon assimilation, RNA-seq

## Abstract

*Polygonatum odoratum* (Mill.) Druce possesses widespread medicinal properties; however, the continuous cropping (CC) often leads to a severe consecutive monoculture problem (CMP), ultimately causing a decline in yield and quality. Photosynthesis is the fundamental process for plant growth development. Improving photosynthesis is one of the most promising approaches to increase plant yields. To better understand how *P. odoratum* leaves undergo photosynthesis in response to CC, this study analyzed the physiochemical indexes and RNA-seq. The physiochemical indexes, such as the content of chlorophyll (chlorophyll a, b, and total chlorophyll), light response curves (LRCs), and photosynthetic parameters (F_v_/F_m_, F_v_/F_0_, F_m_/F_0_, Pi_abs_, ABS/RC, TRo/RC, ETo/RC, and DIo/RC) were all changed in *P. odoratum* under the CC system. Furthermore, 13,798 genes that exhibited differential expression genes (DEGs) were identified in the *P. odoratum* leaves of CC and first cropping (FC) plants. Among them, 7932 unigenes were upregulated, while 5860 unigenes were downregulated. Here, the DEGs encoding proteins associated with photosynthesis and carbon assimilation showed a significant decrease in expression under the CC system, such as the PSII protein complex, PSI protein complex, Cytochorome b6/f complex, the photosynthetic electron transport chain, light-harvesting chlorophyll protein complex, and Calvin cycle, etc., -related gene. This study demonstrates that CC can suppress photosynthesis and carbon mechanism in *P. odoratum*, pinpointing potential ways to enhance photosynthetic efficiency in the CC of plants.

## 1. Introduction

Medicinal plants often suffer from a severe consecutive monoculture problem (CMP), also known as replant disease. CMP of this plant in the same soil reduces production and quality [1,2]. This may be connected to soil nutrient deficits, the increased allelopathic autotoxicity of root exudates, the accumulation of fungal pathogens, and imbalances in the soil microbial community [3]. 

*P. odoratum* (Mill.) Druce is a well-known traditional Chinese herb that belongs to the rhizome plants of the Liliaceous family. Recently, the *P. odoratum* requirement has been increased yearly due to its multiple pharmacological effects, particularly in antibacterial, anti-inflammatory, and neuroprotective activities [4]. To meet the market demand, continuous cropping (CC) has increased dramatically for *P. odoratum*. Because of this, *P. odoratum* is also prone to replanting diseases, leading to a decrease in the quality and yield of *P. odoratum*, and it has become a major problem in parts of China. Plants are known to be exposed to multiple abiotic and biotic stressors, including pathogen invasion, cold, heat, salinity, and CC system, which influence physiological and biochemical changes, thus further affecting plant metabolism simultaneously in nature [5,6]. Most of these physiological and biochemical changes are mainly the photosynthetic performance of plant leaves [7]. Photosynthesis is one of the most sensitive components of photosynthetic organisms to adverse environmental conditions, including the CC system [1], and is crucial for plant metabolism. There are, however, only a few reports that the CC system impacts the photosynthetic process [1], especially in the photosynthesis of *P. odoratum*. The photosynthetic efficiency of leaves relies on the characteristics and amounts of components of the photosynthetic machinery, which is influenced by environmental factors, including nutrient availability [8].

Transcriptome sequencing has also been extensively applied to identify and characterize genes in response to the mechanism of various organisms to different environmental stresses, including the CC system [1,9,10]. According to the RNA-Seq analysis, the gene expression involved in photoprotection and carbon fixation was upregulated against high light, temperature, salt, and UV-B stresses in *Ulva linza* [11]. In addition, transcriptome analysis revealed that a large number of genes responsible for chlorophyll biosynthesis, photosynthesis, carbohydrate metabolism, and ROS-scavenging pathways were highly expressed in *Chlamydomonas reinhardtii* and *Dunaliella. salina* when exposed to methylmercury, Cd, and Cu stress [12,13]. This present study of *P. odoratum* mainly concentrates on the physiology of the CC system. For example, our research discovered that the CC system enhanced the phenolic acid anabolism in *P. odoratum* [14], accumulating phenolic acids in the soil. Additionally, we identified a potential miRNA-mRNA regulatory network involved in the biosynthesis of phenolic acids in the CC of *P. odoratum* roots [15]. Besides this, progressive studies have illustrated recently that the CC system may result in photosynthesis changes. Plant growth, production, and yield are typically influenced by photosynthesis and are usually vulnerable to abiotic stress [5]. Therefore, improving photosynthesis is central to enhancing crop yield. 

However, the associated regulatory mechanisms in CC of *P. odoratum*, especially the regulation of photosynthesis and its mechanism, have received little attention. This study sought to investigate photosynthesis and carbon fixation changes and the regulatory mechanisms. The findings could enhance our understanding of photosynthesis and contribute to identifying potential ways to improve the photosynthetic efficiency in CC of *P. odoratum*.

## 2. Results

### 2.1. Effect of Continuous Cropping on the Photosynthetic Physiological Characteristics in P. odoratum

The chlorophyll content and photosynthetic light response curves (LRC) are important indexes of photosynthesis activity in plants [16]. To investigate the photosynthetic response to the CMP in leaf tissues of *P. odoratum*, we analyzed the photosynthesis-related attributes. CC transcript dramatically decreased chlorophyll contents, which was 31.56% less than the amount in the first cropping (CC, Figure 1A). Similarly, chlorophyll b and total chlorophyll were significantly lower in the CC leaves than in the first cropping (FC) leaves (Figure 1B,C). Moreover, comparisons of leaf photosynthetic light response curve suggested a clear difference in leaf photosynthetic capacity between CC and FC, and this difference was small under low levels of photo-synthetically active radiation (PAR < 300 µmol·m^−2^·s^−1^) but significantly intensified as PAR increased. Moreover, CC significantly inhibits F_v_/F_m_ (Figure 2A), F_v_/F_0_ (Figure 2B), and F_m_/_F0_ (Figure 2C) of *P. odoratum* leaves (by 38.36%, 14.13%, and 30.50% declines, respectively). At the CMP, a significant decrease in PI_Abs_ was observed in CC leaves (Figure 2D). Figure 2E–H displayed phenomenological energy fluxes. The CC significantly reduced ABS/RC (Figure 2E), TRo/RC (Figure 2F), and ETo/RC (Figure 2G) in *P. odoratum* leaves by 36.53%, 22.69%, and 12.283%, respectively. However, there was an observed increase of 179.43% in DIo/RC at CC leaves (Figure 2H). These results suggested that the CC inhibits chlorophyll synthesis and affects the values of chlorophyll fluorescence parameters.

### 2.2. Gene Differential Expressed in Response to Continuous Cropping in P. odoratum

To determine the gene expression patterns in *P. odoratum* leaves when exposed to CMP, we conducted high-throughput sequencing. The databases provided annotations for 435,090 unigens, encompassing NR, NT, KO, SwissProt, PFAM, GO, and KOG. Notably, the GO database alone annotated 176,156 unigenes. 

In leaf tissues of *P. odoratum*, a total of 13,798 DEGs were screened out, including 7932 upregulated DEGs and 5866 downregulated differentially expressed genes (DEGs) (padj < 0.05, and the |log2ratio| ≥ 1; Figure 3A,B). The gene expression patterns were observed by conducting hierarchical clustering of all DEGs and evaluating them using the log_10_ FPKM of the 2 groups (Figure 3C). These results suggest that CC markedly affected the transcription of a subset of genes in response to CMP stress. 

In order to investigate the function of the CC-induced DEGs under CMP stress, the DEGs were used in the Gene Ontology (GO) enrichment analysis. These DEGs were significantly enriched in 857 GO functional items (*p* < 0.05) for the biological process, molecular function, and cellular component categories (Appendix A). The upregulated DEGs were mainly enriched in the metabolic process, the methionine metabolic process, external encapsulating structure, ubiquitin ligase complex, catalytic activity, transferase activity, etc. (Figure 4A); the downregulated DEGs were mainly enriched in the metabolic process, single-organism metabolic process, membrane part, membrane protein, catalytic activity oxidoreductase activity, etc. (Figure 4B). This suggested that CC may result in obvious noticeable alterations in *P. odoratum*. 

Additionally, our previous study revealed that the DEGs of CC vs. FC are mainly related to stilbenoid, brassinosteroid, and phenylpropanoid biosynthesis, phenylalanine metabolism, etc. [17]. In our study, KEGG enrichment analysis with *p* < 0.05 showed that 13,798 DEGs were assigned to 116 KEGG pathways (Appendix A), and the top 10 most enriched pathways were displayed in photosynthesis, photosynthetic antenna proteins, carbon fixation in photosynthetic organisms, stilbenoid, diarylheptanoid, and gingerol biosynthesis, phenylpropanoid biosynthesis, glyoxylate and dicarboxylate metabolism, flavonoid biosynthesis, limonene, and pinene degradation, alpha-Linolenic acid metabolism, and glutathione metabolism, which were a rich factor, number of genes (Appendix A, Figure 4C,D). Moreover, we found that these significantly enriched the upregulated DEGs, mainly phenylpropanoid biosynthesis (189 DEGs), starch and sucrose metabolism (63 DEGs), plant hormone signal transduction (40 DEGs), etc. (Figure 4C). In contrast, the downregulated DEGs were significantly enriched in pathways such as photosynthesis (164 DEGs), carbon fixation in photosynthetic organisms (200 DEGs), glyoxylate and dicarboxylate metabolism (153 DEGs), and photosynthesis−antenna proteins (110 DEGs) (Figure 4D), etc. This indicated that the CC system primarily affected the pathways associated with photosynthesis, carbohydrate metabolism, and phenylpropanoid biosynthesis of *P. odoratum*. 

### 2.3. Transcripts with Increased Levels of Encoded Proteins Involved in Photosynthesis and Carbon Assimilation of P. odoratum

The GO and KEGG enrichment analysis revealed that the CC system significantly induced the down- or upregulation of unigenes encoding proteins relevant to photosynthesis and carbon assimilation. These genes were found to be associated with photosynthesis, antenna proteins, and carbon fixation (Appendix A). To respond to the CC system, the levels of 14 genes encoding the PSII protein complex (PsbA, PsbD, PsbC, PsbB, PsbE, PsbI, PsbO, PsbP, PsbQ, PsbR, PsbS, PsbY, Psb27, and Psb28), as well as the PSI protein complex (PsaA, PsaB, PsaD, PsaE, PsaF, PsaG, PsaH, PsaK, PsaL, PsaN, and PsaO) were downregulated. (Appendix A, Figure 5, Figure 6 and Appendix A). PetB, PetD, and PetC genes encoding the Cytochorome b6/f complex were also downregulated (Appendix A, Figure 5, Figure 6 and Appendix A). Similarly, for the genes involved in the photosynthetic electron transport chain, the expression of PetE, PetF, PetH, and PetJ significantly increased compared with those in FC leaves (Appendix A, Figure 5, Figure 6 and Appendix A). Five lower abundant F-type ATPase, beta, alpha, gamma, deIta, and b had also decreased expression (Appendix A, Figure 5, Figure 6 and Appendix A). Moreover, 12 genes (Lhca l–5 and Lhcb1–7) encoding light-harvesting chlorophyll protein compiexv (LHX) protein were also found to be downregulated (Appendix A, Figure 5 and Figure 6 and Appendix A). Finally, three genes (rpiA, rpe, and rbcS) involved in the carbon fixation in photosynthetic organisms were downregulated, while the maeB gene encoding Malate to CO_2_ was upregulated (Appendix A, Figure 5). These genes played key roles in the process of photosynthetic electron transport and ATP synthesis, which showed that CC could significantly inhibit the growth and photosynthesis of *P. odoratum*. 

### 2.4. qRT-PCR Validation of RNA-Seq Expression Changes

To further validate our RNA-seq results, we selected 20 unigenes for qPCR analysis. These genes were linked to a complex of membrane proteins, the chain of photosynthetic electron transport proteins that act as antennas, and the process of carbon fixation. In photosynthesis, the first step is the absorption of light energy by chlorophyll and its ionization. Four encoding PSII proteins-related genes, namely PsbA (60288.94187), PsbR (60288.8457), PsbO (60288.223377), and PsbY (60288.229913), were selected for the qPCR. In comparison to the FC of leaf samples, the CC of leaf samples notably decreased the expression of the PsII protein genes (Figure 7A). Meanwhile, PsaE (60288.301676) and PsaO (60288.229475), PetB (60288.115236) and PetC (60288.222952), PetF (60288.216724) and PetJ (60288.110783), β-ATPase (60288.373821) and γ-ATPase (60288.241587), and Lhca2 (60288.232211) and Lhca3 (60288.197477) encoding PSI proteins, the cytochrome b6/f complex, photosynthetic electron transport, F-type ATPase, light-harvesting chlorophyll protein complex (LHC), respectively, were also lowly expressed. In addition, glpX-SEBP (60288.220661), rpiA (60288.376230), and rbcS (60288.221750), which encode enzymes for carbon fixation in photosynthetic organisms, were downregulated in CC compared to FC (Figure 7A), whereas the malate dehydrogenase (maeB, 60288.210651) enzymes involved in pyruvate consumption increased (Figure 7A), which indicated that CC inhibited photosynthesis and reduced the carbon fixation. Moreover, the reliability of transcriptome sequencing was validated using a Pearson correlation coefficient of 0.5138 (Figure 7B).

## 3. Discussion

The CC system is prevalent in Chinese medicinal herbs in every cultivation area, severely reducing biomass and crop quality. We also found that CC inhibited the growth, yield, and quality of *P. odoratum* and impacted the synthesis of phenolic acids and its underlying mechanism in *P. odoratum* [14,15]. Besides this, recently, progressive studies have illustrated that CC may result in photosynthesis changes [1]. Plant growth, production, and yield depend on photosynthesis and are usually susceptible to abiotic stress [18]. Therefore, improving photosynthesis is central to enhancing crop yield. However, the associated regulatory mechanisms in CC of *P. odoratum*, especially the underlying molecular mechanisms, have received little attention. 

It is well known that chlorophyll content is a key indicator of the photosynthetic capacity of plants. It is a vector for absorbing and transmitting light energy and is also indispensable in electron transportation as an electron transporter, reflecting plant tolerance to environmental stress [19]. Here, the chlorophyll a and b content in CC of *P. odoratum* leaves was significantly lower than in FC, especially total chlorophyll. This is likely due to the lack of nitrogen and phosphorus in the soil under CC, which inhibits chlorophyll biosynthesis and restricts photosynthesis [20]. Our previous study found that nitrogen and phosphorus in the CC soil were considerably lower compared to the FC soil [14]. These results were consistent with the research reported from the studies performed in the CC of Angelica sinensis by Zhang [2]. LRC is useful in plant physiology for assessing the photosynthetic performance of plants. In our study, CC significantly inhibits LRCs of leaves in *P. odoratum*, indicating CC affects the photosynthetic capacity and efficiency. Additionally, in photosynthesis, the leaf’s chlorophyll can utilize the absorbed light for photochemistry, while any surplus energy can either be released as heat or re-emitted as chlorophyll fluorescence [21]. PI_ABS_ (Performance Index) is a multi-parametric expression that combines the three main steps that precisely regulate photosynthetic activity via a PSII (the absorption of light energy (ABS)), trapping of excitation energy (TR), and conversion of excitation energy to electron transport (ET)) reaction center (RC) complex [22]. Therefore, we utilized it to assess the overall photosynthetic performance of plants under the CC system. In our study, CC significantly reduced ABS/RC, TRo/RC, and ETo/RC in *P. odoratum* leaves, but an increment in DIo/RC was observed in CC leaves. These results demonstrate the strong link between analyzing the photosynthetic activity features in response to the CC system. Moreover, F_v_/F_m_ is used as a chlorophyll fluorescence parameter, describing the maximum photochemical efficiency of PSII, and is a sensitive indicator of photosynthetic performance in plants [23]. According to reports, plants under stress exhibit lower F_v_/F_m_ values, which is accompanied by a reduction in the ability of phytochemicals within the PSII to quench [24]. This study found that CC has been observed to reduce the F_v_/F_m_, F_v_/F_0_, and F_m_/F_0_ in *P. odoratum*. Consistent with this current study’s findings, previous research on vegetables showed that F_v_/F_m_ values were substantially affected by various leafy vegetable cropping systems [25]. 

Photosynthesis is composed of the light and the dark reactions in the chloroplasts. In the light reaction plant, light energy is absorbed and conserved in ATP and NADPH. It is mainly driven by four intrinsic multi-subunit membrane–protein complexes (PSI, PSII, the cytochrome b6/f complex, and ATP synthase) [26]. In plants, PSI can trap light and create PSI-LHCI supercomplexes, which are composed of light-harvesting complex I (LHCI), RC pigments, and the electron transfer system, including PsaA, PsaB, PsaC, PsaD, PsaE, PsaF, PsaH, PsaI, PsaL, PsaN, PsaO and Lhca1–Lhca9 in plants [27]. They participated in light harvesting and transferring electrons from plastocyanin (PC) to ferredoxin (Fd) [28]. Researchers have discovered that certain natural stresses (such as salt and drought) may reduce light-harvesting and photosynthetic activities, PSI and II, cytochrome b6/f, and ATP synthase gene expression levels [29]. In particular, genes encoding components of PSI and II are repressed during natural stresses (such as drought stress) [30]. In this study, the CC system could inhibit the PSI-LHCI supercomplex of *P. odoratum* by downregulating 14 genes (PsaA, PsaB, PsaD, PsaE, PsaF, PsaG, PsaH, PsaK, PsaL, PsaN, and PsaO) and 12 genes (Lhca l–5 and Lhcb1–7) that encode the light-harvesting chlorophyll protein complex (LHX) protein to suppress light-harvesting and electron transfer. Similarly, the PSII family protein, caused by the CC system, indicated that these genes capable of seizing light were significantly suppressed, leading to a slowdown in the electron transfer chain. Additionally, the Cytochrome b6/f complex is recognized for its crucial involvement in the transfer of electrons during photosynthesis, connecting PSI and PSII, and supplying solar energy for the synthesis of ATP [31]. Here, PetB, PetD, and PetC genes encoding the Cytochorome b6/f complex were also downregulated, and the genes involved in the photosynthetic electron transport chain, the expression of PetE, PetF, PetH, and PetJ showed a significant decrease compared with the those in FC leaves, also suggesting the possibility of electron transport inhibition. Subsequently, the light-induced electrons were transferred to F-type ATP synthase within chloroplasts, leading to the synthesis of ATP [32]. In our study, the downregulated expression of beta, alpha, gamma, deIta, and b indicated that ATP synthesis was restrained under the CC system. 

Photosynthesis is the primary pathway for producing carbohydrates essential for cell growth and proliferation. After the light reaction, the dark reaction takes place where the Calvin cycle uses the ATP and NADPH produced from the light reaction to convert CO_2_ into carbohydrates in the stroma of the chloroplast [33]. In this study, some unigenes associated with carbon assimilation exhibited differential expressions following exposure to CC. Among these, the rbcS gene encodes the small subunit of Rubisco, a key enzyme for carbon assimilation, which catalyzes ribulose-1,5-bisphosphate (RuBP) to 3-phosphoglycerate (PG) and plays an important role in CO_2_ fixing and photosynthesis [34]. The CC system significantly downregulated rbcS, whereas upregulated maeB is involved in converting malate into oxaloacetate (using NADP+). This suggests CC may inhibit photosynthesis and suppress the Calvin cycle to produce less glucose.

## 4. Materials and Methods

### 4.1. Plant Materials

Yihong Hu’s research group kindly provided the *P. odoratum* leaves in CC and FC [14]. The *P. odoratum* of CC was cultivated in the same area where the identical plants were collected, while *P. odoratum* of FC grew in fields adjacent to CC where the cabbages had been harvested. Leaf samples from CC and FC plants were randomly collected from the fields during the rhizome expansion stage for RNA-seq and qRT-PCR analysis. Following collection, all samples were promptly placed in liquid nitrogen and preserved at −80 °C for future experiments.

### 4.2. Chlorophyll Content Measurements

The chlorophyll concentration was measured using a 95% ethanol extraction procedure, and CC and FC *P. odoratum* leaves were sliced and immersed in 10 mL 95% ethanol for 48 h in the dark until the leaves turned completely white. The absorbance was measured at 665 and 649, respectively, as described by Lichtenthaler [35]. Chlorophyll concentration was calculated according to the following formula and expressed as mg/g FW of the sample: chlorophyll a (chl a) = 13.95 OD_665_ − 6.88 OD_649_, chlorophyll b (chl b) = 24.96 OD_649_ − 7.32 OD_665_. Additionally, total chlorophyll = chlorophyll a + chlorophyll b.

### 4.3. Chlorophyll Fluorescence Parameters Measurements

Fresh leaves of uniform length were chosen after a 30 min dark incubation. Fv/Fm, F_v_/F_0_, and F_m_/F_0_ of CC and FC *P. odoratum* leaves were calculated by measuring the initial fluorescence (Fo) and maximum fluorescence (Fm) in the light using a chlorophyll fluorometer (FluorPen FP-100, Photon Systems Instruments, Czech Republic). In contrast, variable fluorescence yield (FV) was calculated under dark adaptation. Performance index (PI_ABS_), PSII per reaction center (RC) (ABS/RC, TRo/RC, ETo/RC, and DIo/RC) of CC and FC *P. odoratum* leaves were measured using a chlorophyll fluorometer. Each measurement was repeated three times with ten replications per experiment.

### 4.4. RNA Isolation, Quantification, and Qualification

RNA was isolated from intact leaves of CC and FC *P. odoratum* using the TRIzol reagent (Invitrogen, Carlsbad, CA, USA) following the instructions provided by the manufacturer. The concentration of RNA was determined by employing a NanoDrop 2000 instrument (Thermo Fisher, Waltham, MA, USA), while the integrity of RNA was evaluated using the RNA Nano 6000 Assay Kit (Agilent Technologies, Santa Clara, CA, USA). Three replicates were utilized for mRNA library construction and qRT-PCR verification, each sample containing 1µg of total RNA.

### 4.5. RNA-Sequencing and Analysis

The transcriptome of *P. odoratum* leaves was examined by analyzing two libraries (CC and FC) specifically created for RNA-Seq [36]. Novogene Technology Co., Ltd.,(Beijing, China) prepared and sequenced the library. The Illumina HiSeq 2500 machine was utilized to sequence every library, employing paired-end protocols with a read length of 150 base pairs. Clean reads were obtained using the raw pair-end data. All raw data were uploaded to the NCBI’s Sequence Read Archive (SRA) public database with accession number PRJNA507291. 

Clean reads from mRNA sequencing data were obtained by removing adapters, low-quality reads, and ambiguous were removed. Next, Trinity (version 2.0.6) was utilized to carry out the de novo transcriptome assembly [37]. The longest sequence region obtained with Trinity for each cluster was referred to as a unigene. The unigenes were annotated using NR (NCBI non-redundant protein sequences database), Nt (NCBI nucleotide sequences database), Pfam (Protein family database), KOG (eukaryotic orthologous groups), COG (Clusters of Orthologous Groups), SwissProt (Swiss Institute of Bioinformatics databases), and KEGG (Kyoto Encyclopedia of Genes and Genomes databases). The expression level of each unigene was normalized by calculating the reads per kilobase per million reads (RPKM). The DESeq R package (Version 1.10.1) was utilized to screen the differential expression (n = 3) with gene symbol annotation, applying the criteria of |log2ratio| ≥ 1 and padj < 0.05 [38]. Then, the DEGs underwent KEGG pathway annotation analyses with the utilization of KOBAS (version 2.0). The KEEG pathways with a false discovery rate (FDR) of less than or equal to 0.05 showed a significantly higher enrichment of DEGs.

### 4.6. Quantitative RT-PCR Validation

According to our previous methods, twenty genes were selected for qRT-PCR to verify photosynthesis genes [15]. To validate the level of mRNA expression, a PrimerScript RT Mix (Takara, Beijing, China) was used to generate cDNA from 1 µg total RNA for constructing the mRNA library. qRT-PCR primers were designed using Primer Premier 5.0 software (Premier, Ottawa, ON, Canada). We use the R = 2^^−ΔΔCt^ method to calculate each tested reference gene [39]. The internal reference gene utilized was the 18S gene. Each sample was subjected to three technical replicates. This test includes a list of all the primers provided in Appendix A.

### 4.7. Statistical Analysis 

The data were presented as the average ± SEM. Statistical significance was evaluated using the independent sample *t*-test. Statistical significance was determined using SPSS software packages (version 21.0 for Windows; SPSS Inc., Chicago, IL, USA). Significant variations were observed when *p* < 0.05, with a tendency observed when 0.05 < *p* < 0.1. Graphs were created utilizing GraphPad Prism 5.01 (developed by GraphPad Software, located in La Jolla, CA, USA).

## 5. Conclusions

In summary, KEGG pathway enrichment analysis of DEGs and qRCR analysis confirmed that CC can significantly downregulate the genes relevant to photosynthesis (the PSII, PSI, Cytochrome b6/f complex, photosynthetic electron transport, and F-type ATPase) and carbon mechanism, as well as photosynthetic pigment. Our findings could improve our understanding of the physiological and molecular mechanisms involved in photosynthesis in CC obstacle and provide a theoretical basis (candidate genes) for improving the photosynthetic capacity and yield in continuous cropping obstacle of *P. odoratum*.

## Figures and Tables

**Figure 1 plants-12-03374-f001:**
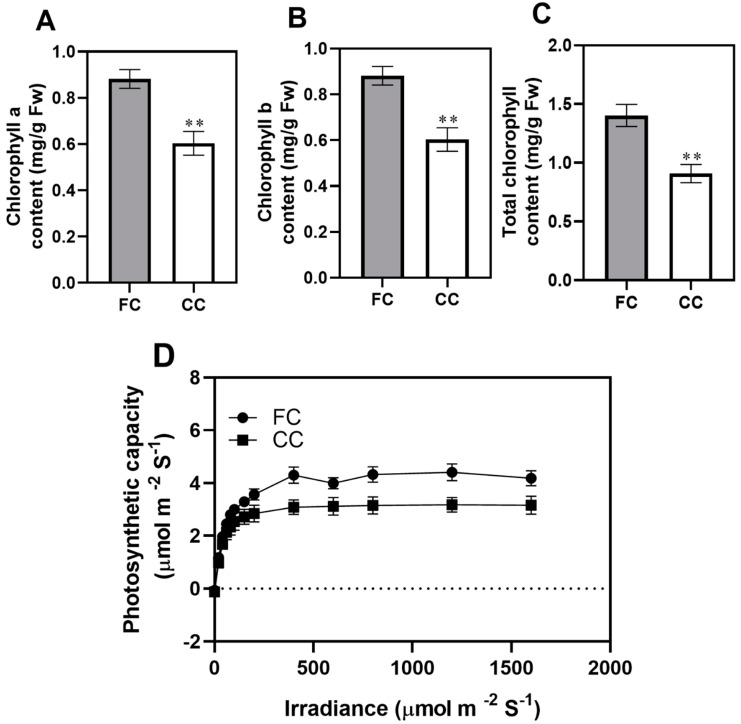
Contents of photosynthetic pigments in FC and CC leaf tissues of *P. odoratum*: (**A**) chlorophyll a, (**B**) chlorophyll b, (**C**) total chlorophyll, and (**D**) photosynthetic light response curves (LRCs). The results are displayed as mean ± standard error of the mean (SEM) (n = 6). ** represent significant at *p* ≤ 0.01 levels according to Student’s *t*-test. FC represents the first cropping leaf samples, while CC represents consecutive cropping leaf samples.

**Figure 2 plants-12-03374-f002:**
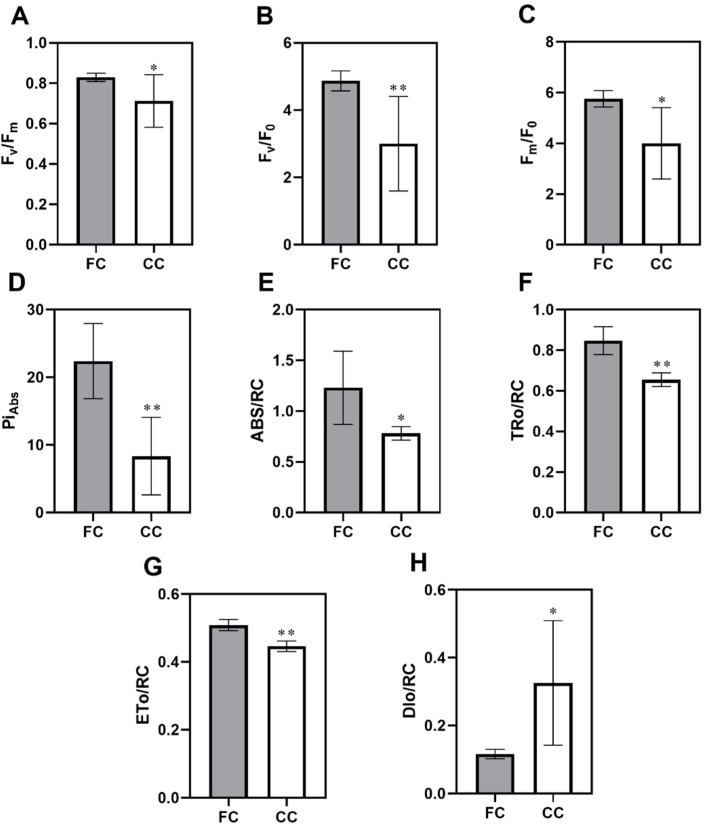
Chlorophyll fluorescence parameters in FC and CC leaves tissues of *P. odoratum*: (**A**) the maximal quantum yield of PSII photochemistry (F_v_/F_m_), (**B**) ratio of maximal and minimal fluorescence yield (F_m_/F_0_), (**C**) the potential photochemical efficiency (F_v_/F_0_), (**D**) the performance index absorbance (Pi_Abs_), (**E**) the absorption flux per reaction center (ABS/RC), (**F**) the energy flux trapped per reaction center (TRo/RC), (**G**) the electron transport flux per reaction center (ETo/RC), and (**H**) the dissipated energy flux per reaction center (DIo/RC). The results are displayed as mean ± SEM (n = 6). *, ** represent significant at *p* < 0.05 and *p* < 0.01, and levels according to Student’s *t*-test, respectively. FC represents the first cropping leaf samples, while CC represents consecutive cropping leaf samples.

**Figure 3 plants-12-03374-f003:**
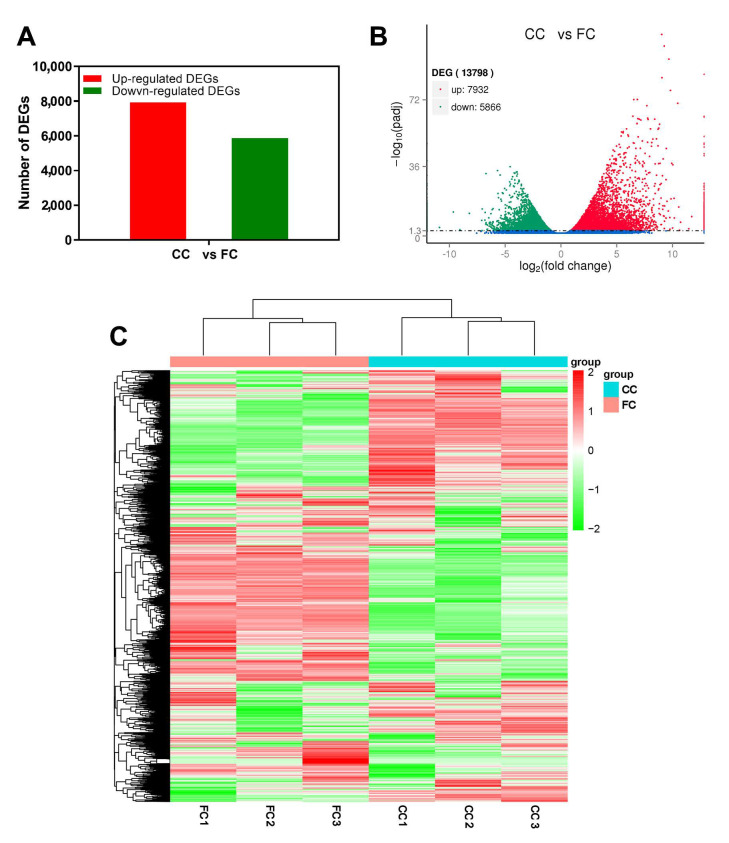
The expression profile of DEGs in CC and FC of *P. odoratum*. (**A**) Number of up− and downregulated DEGs. (**B**) Volcano plot showing the DEGs identified in two different libraries. To determine the significance of DEGs, the threshold q < 0.05 was used. Red and green dots represent upregulated and downregulated genes, respectively. (**C**) Heatmap and clustering analysis of DEGs. FC represents the first cropping leaf samples, while CC represents consecutive cropping leaf samples.

**Figure 4 plants-12-03374-f004:**
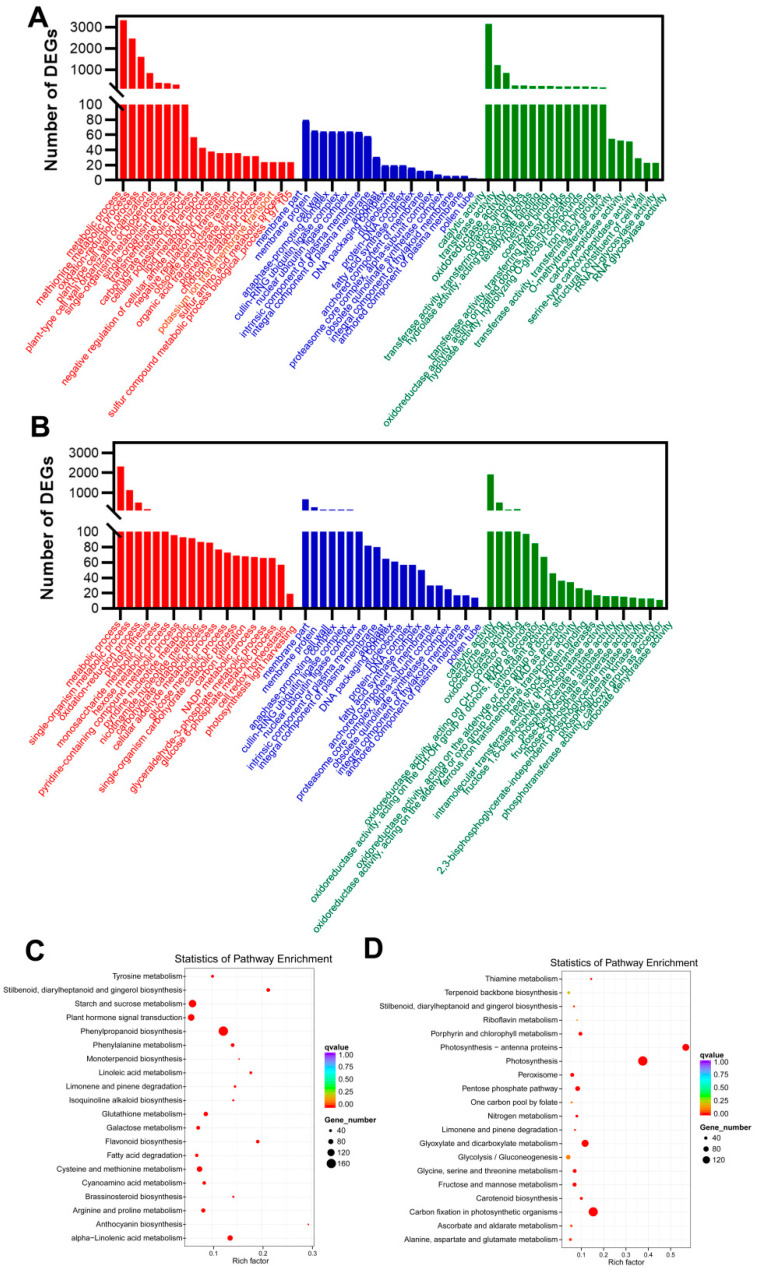
Perform GO and KEGG enrichment analysis on DEGs in *P. odoratum* leaves in response to CMP. DEGs represent differentially expressed genes. FC represents first cropping, and CC represents continuous cropping: (**A**) upregulated DEGs in biological processes, molecular functions, and cellular components, respectively, and (**B**) downregulated DEGs in biological processes, molecular functions, and cellular components, respectively. (**C**) Enrichment of KEGG pathway in upregulated DEGs of CC vs. FC leaf tissues and (**D**) enrichment of KEGG pathway in downregulated DEGs of CC vs. FC leaf tissues.

**Figure 5 plants-12-03374-f005:**
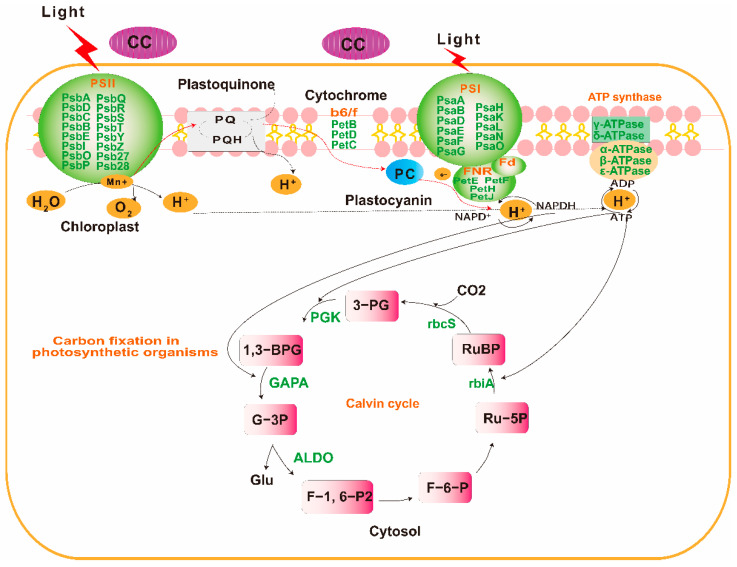
Differential expression genes analysis of photosynthesis pathway in *P. odoratum* leaves in response to CMP. Green color indicates downregulated genes.

**Figure 6 plants-12-03374-f006:**
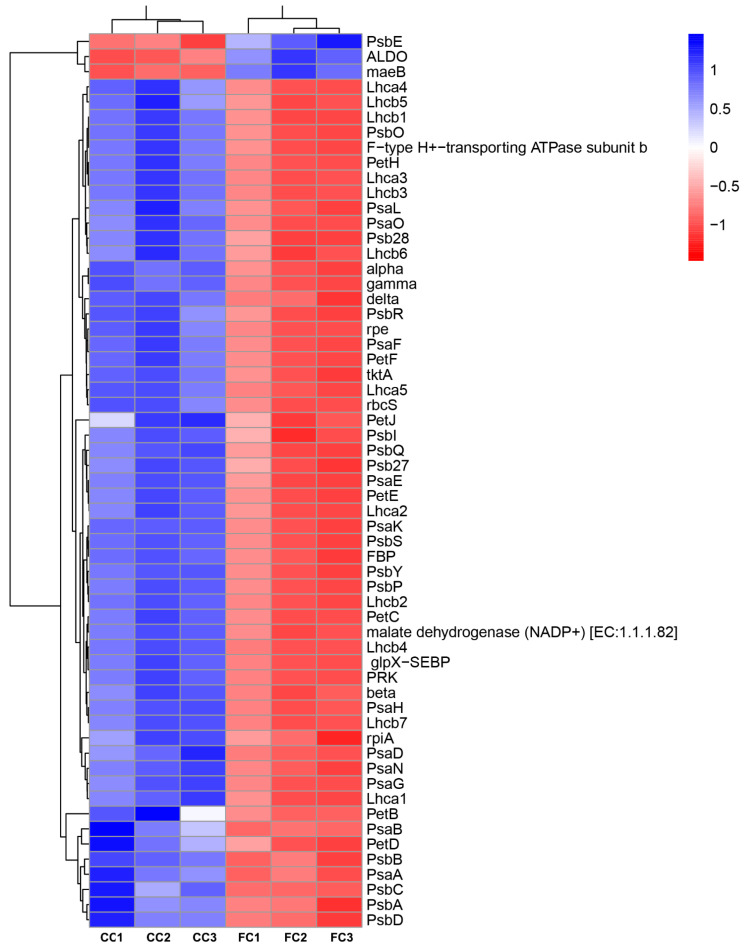
A heatmap displaying differentially expressed genes (DEGs) in CC and FC of *P. odoratum*. FC represents the first cropping leaf samples, while CC represents consecutive cropping’ leaf samples.

**Figure 7 plants-12-03374-f007:**
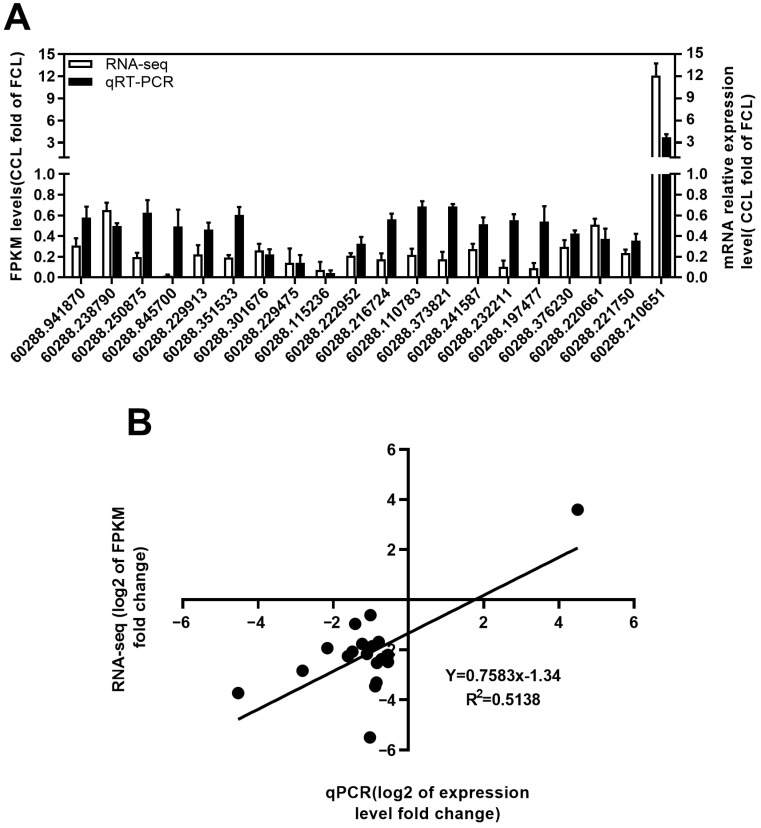
The quality of transcriptome sequencing was verified via qRT-PCR in the leaves of *P. odoratum*. (**A**) RNA-sequencing on transcript levels and qPCR of 20 unigene genes that regulate photosynthesis were verified. (**B**) Scatterplots were generated using the log2 expression ratios from RNA-seq (*y*-axis) and qPCR (*x*-axis).

## Data Availability

Illumina HiSeq generated RNA-Seq reads and the small RNA sequencing datasets are available in NCBI Sequence Read Archive (SRA) for the Bioproject: PRJNA507291.

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
