# Peer review of "Continuous Cropping Inhibits Photosynthesis of Polygonatum odoratum"

_plants, 2023, doi:10.3390/plants12193374_

Round 1

Reviewer 1 Report

 Review on the manuscript “Continuous cropping inhibits Photosynthesis of Polygonatum 2 odoratum”

The study focuses on the problem of reduced productivity as a result of the serious problem of sequential monoculture (CMP), which is characteristic of the long-term cultivation of many medicinal plants. The authors used physiological measurements (chlorophyll content, PSII efficiency) and molecular genetic approaches (RNA-Sequencing, DEGs, qRT-PCR), and showed a decrease in photosynthesis parameters and down-regulation in gene expression associated with photosynthesis and carbon assimilation as a result of  CMP. However, the manuscript contains many sloppiness and errors (see below). And the Discussion, especially its final part, raises many questions and gives the impression that the authors do not understand the processes of photosynthesis well. In this regard, the manuscript requires significant revision.

Comments:

Abstract

Line 24. FC plants - should be decoded

 4. Materials and Methods 310

4.1 Plant Materials

“Yihong Hu's research group kindly provided the P. odoratum leaves in CC and FC …”– CC and FC should be decoded

4.2 Chlorophyll Fluorescence Measurements – how the content of chlorophyll a and chlorophyll b was calculated and in what units it was expressed?

4.3 Photosynthetic Pigment Content Measurements – this paragraph describes the measurement parameters of chlorophyll fluorescence, not pigment content

2. Results

Since the Results section is located earlier than the Materials and Methods section, the authors  should give a transcript of CC and FC at the beginning of Results.

Figure 2. “Chlorophyll fluorescence parameters in FC and CC leaves tissues of P. odoratum …” - the figure caption must be self-sufficient, therefore it is necessary to give a transcript of FC and CC. And most importantly, the caption doesn't match the figure.

Figure 3. “The expression profile of DEGs in the leaves tissues of P. odoratum comparing CC and FC. 129 (A) Nubmer of up-and down-regulated DEGs. FCL represents the leaf samples from the first crop-130 ping, while represents leaf samples from consecutive cropping. ,// ? CCL ???... -  why such a difference in notation?

Figure 5. “The DEGs and proteins to photosynthesis are associated with the pathway related to photosynthesis. Red and yellow color indicate up- and down-regulated proteins, respectively.”    - not the correct figure caption. An incomprehensible figure - it is not clear where the DEG is, where the proteins are, where is red, where is yellow. Why are the color indicate  up- and down-regulation not the same as in other figures (Red and green)?

Figure 6. A heatmap displaying differentially expressed genes (DEGs). FCL represents the leaf samples from the first cropping, while ?????? represents leaf samples from consecutive cropping.

2.4 qRT-PCR validation of RNA-Seq expression changes – the designations FCL and CCL are used, and not FC and CC as in the rest of the text

217-221 “In addition, glpX-SEBP (60288.220661), rpiA 217 (60288.376230), and rbcS (60288.221750), which encode enzymes that carbon fixation in  photosynthetic organisms, were down-regulated in CCL compared to FCL (Figure 7A), whereas the malate dehydrogenase (maeB, 60288.22175) enzymes involved in pyruvate consumption decreased (Figure 7A),…” – this suggestion is not clear. And what about maeB 60288.210651 - upregulated in the figure and in the appendix table - highlighted in yellow? In line 195 - while the maeB gene encoding Malate to CO2 was upregulated (Table S4, Figure 5)

Discussion

238- 240   “This study sought to investigate photosynthesis and carbon fixation changes and the regulatory mechanisms. The findings have the  potential to enhance our comprehension of photosynthesis and aid in identifying possible  methods to enhance the efficiency of photosynthesis in CC of P. odoratum.” – it's already in the Introduction

279-282 – t's a repetition of 275-278 lines

283 - In this study, the CC system could activate the PSI LHCI supercomplex of P. odoratum by downregulating 14 genes…    How can downregulating of PSI and LHCI genes to activate the protein PSI LHCI supercomplex?

305 “Among these, the rbcS gene is an essential enzyme for carbon assimilation…” - rbcS gene is not a protein - this gene encodes the small subunit of Rubisco, Rubisco is essential enzyme for carbon assimilation.

308 maeB is involved in converting malate into oxaloacetate (using NAD). – in Table S4 there are  maeB: malate dehydrogenase (oxaloacetate-decarboxylating)(NADP+) [EC:1.1.1.40] and  malate dehydrogenase (NADP+) [EC:1.1.1.82] – what do the authors mean?

308-309 – “This suggests CC may inhibit photosynthesis and activate the Calvin cycle to produce more glucose.” - how can CC simultaneously inhibit photosynthesis and activate the Calvin cycle, which is part of photosynthesis (where the main enzyme is Rubisco and rbcS was down-regulated)?

Author Response

Reviewer 1

Open Review

(x) I would not like to sign my review report

( ) I would like to sign my review report

Quality of English Language

(x) I am not qualified to assess the quality of English in this paper

( ) English very difficult to understand/incomprehensible

( ) Extensive editing of English language required

( ) Moderate editing of English language required

( ) Minor editing of English language required

( ) English language fine. No issues detected

Yes Can be improved Must be improved      Not applicable

Does the introduction provide sufficient background and include all relevant references?

(x)   ( )    ( )    ( )

Are all the cited references relevant to the research?

(x)   ( )    ( )    ( )

Is the research design appropriate?

(x)   ( )    ( )    ( )

Are the methods adequately described?

( )    (x)   ( )    ( )

Are the results clearly presented?

( )    ( )    (x)   ( )

Are the conclusions supported by the results?

( )    (x)   ( )    ( )

Comments and Suggestions for Authors

 Review on the manuscript “Continuous cropping inhibits Photosynthesis of Polygonatum 2 odoratum”

The study focuses on the problem of reduced productivity as a result of the serious problem of sequential monoculture (CMP), which is characteristic of the long-term cultivation of many medicinal plants. The authors used physiological measurements (chlorophyll content, PSII efficiency) and molecular genetic approaches (RNA-Sequencing, DEGs, qRT-PCR), and showed a decrease in photosynthesis parameters and down-regulation in gene expression associated with photosynthesis and carbon assimilation as a result of  CMP. However, the manuscript contains many sloppiness and errors (see below). And the Discussion, especially its final part, raises many questions and gives the impression that the authors do not understand the processes of photosynthesis well. In this regard, the manuscript requires significant revision.

Comments:

Abstract

Line 24. FC plants - should be decoded

au: Thanks for your kind suggestion. FC represents the first cropping. It is explained in "2.1" of the “Results “section.

  1. Materials and Methods 310

4.1 Plant Materials

“Yihong Hu's research group kindly provided the P. odoratum leaves in CC and FC …”– CC and FC should be decoded

au: Thanks for your kind suggestion. FC represents the first cropping, while CC represents the continuous cropping. They are receptively explained in the "Introduction" and "Results" sections.

4.2 Chlorophyll Fluorescence Measurements – how the content of chlorophyll a and chlorophyll b was calculated and in what units it was expressed?

au: Chlorophyll concentration was calculated according to the following formula and expressed as mg/g FW of the sample: chlorophyll a (chl a) = 13.95 OD665-6.88 OD649, chlorophyll b (chl b)  = 24.96 OD649-7.32 OD665. Additionally, Total chlorophyll = chlorophyll a + chlorophyll b.

4.3 Photosynthetic Pigment Content Measurements – this paragraph describes the measurement parameters of chlorophyll fluorescence, not pigment content

au: Thanks for your kind suggestion. It has been revised and marked in yellow accordingly.

  1. Results

Since the Results section is located earlier than the Materials and Methods section, the authors  should give a transcript of CC and FC at the beginning of Results.

au: Thanks for your kind suggestion. FC represents the first cropping, while CC represents the continuous cropping. They are receptively explained in the "Introduction" and "Results" sections.

Figure 2. “Chlorophyll fluorescence parameters in FC and CC leaves tissues of P. odoratum …” - the figure caption must be self-sufficient, therefore it is necessary to give a transcript of FC and CC. And most importantly, the caption doesn't match the figure.

au: Thanks for your kind suggestion. You are right. We have added the explanation for FC and CC in the caption of Figure 2. FC represents the first cropping, while CC represents the continuous cropping.

Figure 3. “The expression profile of DEGs in the leaves tissues of P. odoratum comparing CC and FC. 129 (A) Nubmer of up-and down-regulated DEGs. FCL represents the leaf samples from the first crop-130 ping, while represents leaf samples from consecutive cropping. ,// ? CCL ???... -  why such a difference in notation?

au: Thanks. We have revised it as followers: Figure 3. The expression profile of DEGs in CCL and FCL of P. odoratum. We have added explantion for the CCL and FCL accordingly, and it was marked yellow.

Figure 5. “The DEGs and proteins to photosynthesis are associated with the pathway related to photosynthesis. Red and yellow color indicate up- and down-regulated proteins, respectively.”    - not the correct figure caption. An incomprehensible figure - it is not clear where the DEG is, where the proteins are, where is red, where is yellow. Why are the color indicate  up- and down-regulation not the same as in other figures (Red and green)?

au: We would like to make an apology for this mistake. We have corrected this mistake as follows: Figure 5. Differential expression genes analysis of photosynthesis pathway in P. odoratum leaves in response to CMP. The green color indicates down-regulated genes. Thanks a lot!

Figure 6. A heatmap displaying differentially expressed genes (DEGs). FCL represents the leaf samples from the first cropping, while ?????? represents leaf samples from consecutive cropping.

au: We have revised it as followers: Figure 3. The expression profile of DEGs in CCL and FCL of P. odoratum. We have added an explantion for the CCL and FCL accordingly and it was marked yellow. Thanks a lot!

2.4 qRT-PCR validation of RNA-Seq expression changes – the designations FCL and CCL are used, and not FC and CC as in the rest of the text

au: We have added the explanation for FCL and CCL in “2.4” of the “results” section as follows: In comparison to the FC of leaf samples (FCL), the CC of leaf samples (CCL)

217-221 “In addition, glpX-SEBP (60288.220661), rpiA 217 (60288.376230), and rbcS (60288.221750), which encode enzymes that carbon fixation in  photosynthetic organisms, were down-regulated in CCL compared to FCL (Figure 7A), whereas the malate dehydrogenase (maeB, 60288.22175) enzymes involved in pyruvate consumption decreased (Figure 7A),…” – this suggestion is not clear. And what about maeB 60288.210651 - upregulated in the figure and in the appendix table - highlighted in yellow? In line 195 - while the maeB gene encoding Malate to CO2 was upregulated (Table S4, Figure 5)

au: We apologize for this mistake in the maeB unigene number. We have corrected this mistake as follows: whereas the malate dehydrogenase (maeB, 60288.210651) enzymes involved in pyruvate consumption increased (Figure 7A).

Discussion

238- 240   “This study sought to investigate photosynthesis and carbon fixation changes and the regulatory mechanisms. The findings have the  potential to enhance our comprehension of photosynthesis and aid in identifying possible  methods to enhance the efficiency of photosynthesis in CC of P. odoratum.” – it's already in the Introduction

au: Thanks for your kind suggestion. It has been revised and marked in yellow accordingly.

279-282 – t's a repetition of 275-278 lines

au: We apologize for this mistake. We have deleted the superfluous part.

283 - In this study, the CC system could activate the PSI LHCI supercomplex of P. odoratum by downregulating 14 genes…    How can downregulating of PSI and LHCI genes to activate the protein PSI LHCI supercomplex?

Au: We apologize for this mistake. We have revised it and added some discussion as follows: Researchers have discovered that certain natural stresses (such as salt and drought) may reduce light-harvesting and photosynthetic activities, PSI and II, cytochrome b6/f, and ATP synthase gene expression levels (Tang et al, 2018). In particular, genes encoding components of PSI and II are repressed during natural stresses (such as drought stress)( Hayano-Kanashiro et al, 2009). In this study, the CC system could inhibit the PSI-LHCI supercomplex of P. odoratum by downregulating 14 genes (PsaA, PsaB, PsaD, PsaE, PsaF, PsaG, PsaH, PsaK, PsaL, PsaN, and PsaO) and 12 genes (Lhca l-5 and Lhcb1-7) that encode the light-harvesting chlorophyll protein complex (LHX) protein to suppress light-harvesting and electron transfer.

305 “Among these, the rbcS gene is an essential enzyme for carbon assimilation…” - rbcS gene is not a protein - this gene encodes the small subunit of Rubisco, Rubisco is essential enzyme for carbon assimilation.

au: Thanks for your kind suggestion. We have revised it as follows: the rbcS gene encodes the small subunit of Rubisco, a keyrate-limiting enzyme for carbon assimilation, which catalyzes ribulose-1,5-bisphosphate (RuBP) to 3-phosphoglycerate (PG) and plays an important role in CO2 fixing and photosynthesis.

308 maeB is involved in converting malate into oxaloacetate (using NAD). – in Table S4 there are  maeB: malate dehydrogenase (oxaloacetate-decarboxylating)(NADP+) [EC:1.1.1.40] and  malate dehydrogenase (NADP+) [EC:1.1.1.82] – what do the authors mean?

au: we really appreciate your comments. We have corrected accordingly. We used " NADP+" instead of " NAD ".

308-309 – “This suggests CC may inhibit photosynthesis and activate the Calvin cycle to produce more glucose.” - how can CC simultaneously inhibit photosynthesis and activate the Calvin cycle, which is part of photosynthesis (where the main enzyme is Rubisco and rbcS was down-regulated)?

Au: We apologize for this mistake our careless. We have changed this sentence as follows: The CC system significantly down-regulated rbcS, whereas up-regulated maeB is involved in converting malate into oxaloacetate (using NADP+). This suggests CC may inhibit photosynthesis and suppress the Calvin cycle to produce less glucose.

Submission Date

25 August 2023

Date of this review

07 Sep 2023 12:30:40

Reviewer 2 Report

The study was well carried out and the information in general is appropriately presented. In my opinion, it deserves publication after some corrections (see the attached file). Pay special attention to the comments on material and methods where the major shortcomings are to be found.

The English is fine and only minor corrections are required

Author Response

Reviewer 2

Open Review

( ) I would not like to sign my review report

(x) I would like to sign my review report

Quality of English Language

( ) I am not qualified to assess the quality of English in this paper

( ) English very difficult to understand/incomprehensible

( ) Extensive editing of English language required

( ) Moderate editing of English language required

(x) Minor editing of English language required

( ) English language fine. No issues detected

Yes Can be improved      Must be improved    Not applicable

Does the introduction provide sufficient background and include all relevant references?

(x)   ( )    ( )    ( )

Are all the cited references relevant to the research?

(x)   ( )    ( )    ( )

Is the research design appropriate?

(x)   ( )    ( )    ( )

Are the methods adequately described?

( )    ( )    (x)   ( )

Are the results clearly presented?

( )    (x)   ( )    ( )

Are the conclusions supported by the results?

(x)   ( )    ( )    ( )

Comments and Suggestions for Authors

The study was well carried out and the information in general is appropriately presented. In my opinion, it deserves publication after some corrections (see the attached file). Pay special attention to the comments on material and methods where the major shortcomings are to be found.

Comments on the Quality of English Language

The English is fine and only minor corrections are required

au: We really appreciate your comments. We have revised the manuscript by accounting for the helpful comments and requirements from you and it was marked in yellow. Thanks again.

Submission Date

25 August 2023

Date of this review

02 Sep 2023 11:04:30[3, 4]

Round 2

Reviewer 1 Report

Review_2

 In Results - 86 line  - “CC dramatically decreased… “  - “СС” transcript should be added

 Figure 2. …….. And most importantly, the caption doesn't match the figure.  -  this was not taken into account..– the captions to sub-figures D-H are not correct

 Figure 3. “The expression profile of DEGs in the leaves tissues of P. odoratum comparing CC and FC" the authors changed to “The expression profile of DEGs in CCL and FCL of P. Odoratum”

If the authors want to use CCL and FCL, then they must use CCL and FCL throughout the text and in all figures.

 Designations must be the same throughout the text and in all figures.

 238- 240   - The authors made minor edits - these sentences remain a repetition of what is in the introduction. Needs to be rephrased or removed

 307 “….Rubisco, a key rate-limiting enzyme for carbon assimilation…” -   Rubisco is a key enzyme for carbon assimilation. It is better to remove the word “rate-limiting”, since the rate-limiting enzyme for carbon assimilation is currently considered to be the Rubisco-activase enzyme

Author Response

Dear Editor,

Thank you for your handling our manuscript and providing us enough revision time. Enclosed is the revised manuscript (ID: plants-2602378) entitled “Continuous cropping inhibits Photosynthesis of Polygonatum odoratum” by Yan Wang et al.

We have revised the manuscript by accounting for the helpful comments and requirements from you and the reviewers. We would like to detail our point-to-point responses to your and reviewers' comments as follows:

Review_2

 In Results - 86 line  - “CC dramatically decreased… “  - “СС” transcript should be added

au: Thanks for your kind suggestion. CC transcript have been added and marked in yellow accordingly.

 Figure 2. …….. And most importantly, the caption doesn't match the figure.  -  this was not taken into account..– the captions to sub-figures D-H are not correct

au: We apologize for this mistake. We have corrected this mistake as follows:(D) the performance index absorbance (PiAbs), (E) the absorption flux per reaction center (ABS/RC), (F) the energy flux trapped per reaction center (TRo/RC), (G) the electron transport flux per reaction center (ETo/RC), (H) The dissipated energy flux per reaction center (DIo/RC).

 Figure 3. “The expression profile of DEGs in the leaves tissues of P. odoratum comparing CC and FC" the authors changed to “The expression profile of DEGs in CCL and FCL of P. Odoratum”

If the authors want to use CCL and FCL, then they must use CCL and FCL throughout the text and in all figures.Designations must be the same throughout the text and in all figures.

au:Thanks for your kind suggestion. You are right. It has been revised and marked in yellow accordingly.

 238- 240   - The authors made minor edits - these sentences remain a repetition of what is in the introduction. Needs to be rephrased or removed

au:Thanks for your kind suggestion. You are right. We have deleted the setentse “238- 240” and marked it in yellow accordingly.

 307 “….Rubisco, a key rate-limiting enzyme for carbon assimilation…” -   Rubisco is a key enzyme for carbon assimilation. It is better to remove the word “rate-limiting”, since the rate-limiting enzyme for carbon assimilation is currently considered to be the Rubisco-activase enzyme

au: Thanks for your kind suggestion. “….Rubisco, a key rate-limiting enzyme for carbon assimilation…” has been changed into “Rubisco is a key enzyme for carbon assimilation”.